# Effects of Replacing Fishmeal with American Cockroach Residue on the Growth Performance, Metabolism, Intestinal Morphology, and Antioxidant Capacity of Juvenile *Cyprinus carpio*

**DOI:** 10.3390/ani14243632

**Published:** 2024-12-17

**Authors:** Xiaofang Zou, Chenggui Zhang, Bingyan Guo, Yu Cao, Yongshou Yang, Peiyun Xiao, Xiaowen Long

**Affiliations:** 1College of Agriculture and Biological Science, Dali University, Dali 671003, China; 19723605017@163.com (X.Z.); 18287926266@163.com (B.G.); 18187082968@163.com (Y.C.); 2Yunnan Provincial Key Laboratory of Entomological Biopharmaceutical R&D, Dali University, Dali 671000, China; chenggui_zcg@hotmail.com (C.Z.); yysh2257415@126.com (Y.Y.); xpy990120@126.com (P.X.); 3National-Local Joint Engineering Research Center of Entomoceutics, Dali University, Dali 671000, China

**Keywords:** fishmeal replacement, American cockroach, growth performance, metabolism, intestinal morphology, antioxidant capacity, juvenile *Cyprinus carpio*

## Abstract

This study evaluated the potential of substituting fishmeal with American cockroach residue in the feed of juvenile *Cyprinus carpio*. Replacing up to 40% of fishmeal with American cockroach residue significantly improved growth performance, antioxidant capacity, and immune function in juvenile *C. carpio*. However, replacing more than 40% of fishmeal may adversely affect intestinal health. Therefore, determining the optimal replacement ratio is essential for maximizing growth and well-being in juvenile *C. carpio*.

## 1. Introduction

Fishmeal has long served as a fundamental component of aquaculture feed due to its balanced nutrient profile, high palatability, efficient digestibility, and low levels of anti-nutritional factors [1]. However, the decline in marine fishery resources and the impacts of climate change have reduced fishmeal production and led to continuous price increases [2]. To support the sustainability of aquaculture, the search for high-quality alternative protein sources has become a critical area of research [3]. Studies have shown that some protein sources can replace fishmeal without negatively affecting fish growth or feed conversion rates. For example, soybean meal is used for bighead carp [4], cottonseed meal for catfish [5], meat and bone meal for Japanese flounder [6], and poultry feed for black sea bream [7]. However, traditional alternatives often face challenges, such as anti-nutritional factors, low palatability, and imbalanced amino acid profiles, limiting their effectiveness as fishmeal substitutes [8,9].

Thus, developing novel protein sources has become essential. In recent years, insect meal has gained attention as a fishmeal alternative due to its protein content and amino acid protein, which closely resembles fishmeal and is a natural food source for many fish species. Popular insect protein sources currently studied in aquaculture feeds include black soldier fly (*Hermetia illucens*) [10], yellow mealworm (*Tenebrio molitor*) [11], silkworm pupae (*Bombyx mori*) [12], and cricket (*Acheta domesticus*) [13]. Despite these advances, a research gap remains in understanding the potential of alternative insect-based protein sources that are widely available but underutilized. Among these, the American cockroach (*Periplaneta americana*) has recently emerged as a unique alternative. Compared to other insects, American cockroach has a high reproductive capacity in urban environments, is widely available as a byproduct of pest management, and thus holds significant potential as a sustainable protein source [14,15]. However, existing research has primarily focused on traditional insect proteins, such as black soldier fly and yellow mealworm, leaving a gap in comprehensive studies on the nutritional value, sustainability, and applicability of American cockroach residue in aquaculture. Preliminary studies suggest that American cockroach residue offers superior nutritional qualities over some traditional insect proteins, with improved amino acid profiles and fatty acid ratios, suggesting it could serve as a high-quality fishmeal substitute [15,16].

Unlike insects such as black soldier fly [17] and yellow mealworm [18] the American cockroach not only resembles fishmeal in terms of protein content and amino acid composition but is also rich in bioactive components, including lipids, peptides, amino acids, and proteins, contributing to tissue repair, anti-tumor properties, antibacterial activity, and immune enhancement [19,20,21]. The residue of the American cockroach, often discarded or incinerated after active ingredient extraction, contributes to environmental pollution and resource waste [22]. Due to its similarity to fishmeal in protein content and amino acid composition [14] and studies showing that adding cockroach residue does not negatively impact the growth and immune function of species like *Mus musculus*, three-yellow chickens, and Nile tilapia [14,16,23], American cockroach residue shows promise as a sustainable fishmeal alternative for aquaculture, potentially benefiting both environmental conservation and resource utilization.

*Cyprinus carpio*, a freshwater fish of the order Cypriniformes, family Cyprinidae, and genus *Cyprinus* [24], is one of the world’s leading aquaculture species [25] due to its strong environmental adaptability [26], rapid growth, high survival rate, ease of capture, flavorful meat, and rich nutritional content [27]. Studies on fishmeal replacement with insect meal in carp feed have shown improved growth performance, antioxidant capacity, and immune function [28,29]. This study seeks to fill the identified research gap by examining the effects of substituting fishmeal with American cockroach residue on the growth, physiological metabolism, intestinal morphology, and antioxidant capacity of juvenile *Cyprinus carpio*. The findings aim to provide new insights into the potential of underutilized insect proteins and offer practical guidance for the sustainable application of American cockroach residue in aquaculture feed.

## 2. Materials and Methods

### 2.1. Ethics

All experimental protocols were approved by the Animal Welfare and Ethical Committee of Dali University (No. 2020-PZ-46). Fish were maintained in well-aerated water and anesthetized with eugenol (30 mg/L) before sampling. Viscera extraction procedures followed the Guidelines for the Care and Use of Laboratory Animals in China.

### 2.2. Diet Formulation

A basal diet was formulated with fishmeal, soybean meal, and corn protein concentrates as the primary protein sources, and fish oil and soybean oil as the primary fat sources. Five isonitrogenous and isolipidic diets were then prepared by replacing 0%, 20%, 40%, 60%, and 80% of the fishmeal in the basal diet with American cockroach residue (defined as Diet 1, Diet 2, Diet 3, Diet 4, and Diet 5, respectively). The American cockroach residue used in this study was purchased from Yunnan Tengyao Pharmaceutical Co., Ltd (Baoshan, China). All feed ingredients were ground and passed through a 60-mesh sieve before being formulated according to the proportions in Table 1. Vitamins and minerals were added using a stepwise method, followed by the incorporation of soybean oil, fish oil, and water. After thorough mixing, the mixture was pelletized into 1.5 mm diameter pellets, air-dried, packaged in self-sealing bags, and stored at −20 °C for future use.

### 2.3. Experimental Design and Rearing Management

The experiment was conducted at the Yunnan Provincial Department of Education Pharmaceutical Resource Comprehensive Utilization Engineering Research Center Dacang Research Base, Dali, China. Three hundred juvenile *C. carpio* with uniform size, no external injuries, good vitality, and an initial body weight of approximately 74 g were selected for the study. After a 5 min immersion in a 3% saline solution, the fish were randomly distributed into 15 circular canvas buckets (1 m diameter × 1 m height). Following a one-week acclimation period, the formal experiment began, during which the juveniles were fed a basal diet. The study used a five-group design with three replicates per group, and each replicate contained 20 fish. Daily records of feed intake, mortality, and water temperature were kept throughout the experiment. The buckets were continuously aerated using air pumps (24 h), and water changes were made based on real-time monitoring of water quality; water quality parameters (such as pH, dissolved oxygen, nitrite, ammonia nitrogen, etc.) were measured every two days during the experiment using water quality analysis kits. Natural light was used for rearing, and the water temperature was maintained between 21–27 °C. Dissolved oxygen levels exceeded 5.0 mg/L, pH ranged from 7.0 to 8.5, ammonia nitrogen concentration was kept below 0.5 mg/L, and nitrite concentration below 0.15 mg/L. Fish were fed to satiation twice daily at 8:00 and 17:00. Uneaten feed was collected 1 h after each feeding via siphoning, and the dry matter content was recorded to determine feed consumption. Feed amounts ranged from 3% to 5% of the fish’s body weight. The rearing experiment lasted for 10 weeks.

### 2.4. Sample Collection

After the rearing trial, the fish were fasted for 24 h, and the number of surviving fish in each group was recorded. Five fish were randomly selected from each bucket, anesthetized with eugenol (30 mg/L), and their body surface moisture was removed using gauze. Growth indicators were subsequently calculated. Blood samples were drawn from five fish per replicate via the caudal vein using a syringe. The blood samples were stored at 4 °C for 4 h and then centrifuged at 3500 rpm for 5 min. The serum was collected and stored at −40 °C for later analysis of metabolic indicators, antioxidant capacity, and immune parameters. 

Following blood collection, the fish were aseptically dissected on ice. The liver and intestines were collected from each replicate, with intestinal contents gently squeezed out and surface fat removed. In this study, a total of 75 midgut samples were collected for histological analysis, with 5 samples randomly taken from each bucket, resulting in 15 samples per feed group. Specifically, approximately 0.50 cm of midgut was fixed in 4% paraformaldehyde at room temperature for histological analysis. The remaining intestinal tissues and liver were placed in ziplock bags and stored at −40 °C for subsequent analysis of digestive enzyme activity. 

The following formulas were used to calculate survival rate (SR), weight gain rate (WGR), specific growth rate (SGR), feed conversion ratio (FCR), protein efficiency ratio (PER), condition factor (CF), viscerosomatic index (VSI), and hepatosomatic index (HSI):$$SR\;\left(\%\right)=100\times \frac{N_{t}}{N_{0}}$$
$$WGR\;\left(\%\right)=100\times \frac{W_{t}-W_{0}}{W_{0}}$$
$$SGR\;(\%/d)=100\times \frac{lnW_{t}-lnW_{0}}{t}$$
$$FCR=\frac{W_{f}}{W_{t}-W_{0}}$$
$$PER\;(\%)=100\times \frac{W_{t}-W_{0}}{W_{f}\times W_{p}}$$
$$CF\;(g/{cm}^{3}\%)=100\times \frac{W_{t}}{L^{3}}$$
$$VSI\;(\%)=100\times \frac{W_{v}}{W_{t}}$$
$$HSI\;(\%)=100\times \frac{W_{l}}{W_{t}}$$
where *N*_0_ is the initial number of fish, *N_t_* is the final number of fish, *W*_0_ is the initial average body weight, *W_t_* is the final average body weight, *t* is the experimental duration in days, *W_f_* is the total feed intake, *W_P_* is the crude protein content of the feed, *L* is the final average body length, *W_v_* is the average viscera weight at the end of the experiment, and *W_l_* is the average liver weight at the end of the experiment.

### 2.5. Proximate Composition and Amino Acid Analysis

The proximate composition (moisture, crude protein, crude fat, and ash content) of the experimental diets and whole fish was assessed using standard methods outlined by the Association of Official Analytical Chemists [30]. Crude protein content was determined by the Kjeldahl method, crude fat content was measured using the Soxhlet extraction method with petroleum ether as the solvent, crude ash content was established by ashing in a muffle furnace at 550 °C for 8 h, and moisture content was evaluated using the constant temperature drying method at 105 °C. The amino acid content in the experimental diets and whole fish was analyzed following the method described by Blackburn (1968) [31]. Methionine content was determined using a formic acid oxidation hydrolysis method. The proximate composition and amino acid content of the experimental diets are presented in Table 2.

### 2.6. Determination of Antioxidant and Immune Indices

Samples from the same bucket of liver or intestine were combined into a single sample, respectively. Approximately 0.5 g liver or intestine was homogenized in 4.5 mL of cold 0.85% saline solution using an IKA T10B homogenizer (IKA, Staufen, Germany) for 1 min. Subsequently, a 10% aliquot of the homogenate was centrifuged at 12,000 rpm and 4 °C for 10 min using an Eppendorf centrifuge (5417R, Eppendorf, Hamburg, Germany). The supernatant was collected and stored at −40 °C for subsequent analysis. The following parameters were measured in the tissue homogenate supernatant and serum using a spectrophotometer (T6, Beijing Pukenye General Instrument Co., Ltd., Beijing, China) and commercially available kits (Nanjing Jiancheng Bioengineering Institute, Nanjing, China): trypsin (Cat.A080-2), lipase (Cat.A054-1), α-amylase (Cat.C016-1), total soluble protein (TP, Cat.A045-2), alanine aminotransferase (ALT, Cat.C009-1), aspartate aminotransferase (AST, Cat.C010-1), total amino acids (T-AAs, Cat.A026-1), blood urea nitrogen (BUN, Cat.C013-2), total cholesterol (TC, Cat.A111-2), triglycerides (TGs, Cat.A110-2), glucose (GLU, Cat.F006-1), total antioxidant capacity (T-AOC, Cat.A015-1), total superoxide dismutase (T-SOD, Cat.A001-1), alkaline phosphatase (ALP, Cat.A059-1), malondialdehyde (MDA, Cat.A003-1), acid phosphatase (ACP, Cat.A060-1), and immunoglobulin M (IgM, Cat.H109-1).

### 2.7. Midgut Histology Analysis

After fixation with 4% paraformaldehyde, the midgut was excised, rinsed with water, dehydrated using graded ethanol, cleared with xylene, embedded in paraffin, and sectioned to a thickness of 5 μm. Hematoxylin–eosin (HE) staining was performed, and the sections were sealed with neutral gum. Once the samples dried, an Olympus microscope was used to observe and photograph the crypt depth, villus length, and muscle thickness. The histological images were subsequently analyzed and processed using K-Viewer v1.5.3 software.

### 2.8. Statistical Analysis

Data are presented as means ± standard error (SE). Levene’s test was used to assess homogeneity of variances. Where necessary, data were transformed using arcsine square root or logarithmic transformations prior to analysis. A one-way analysis of variance (ANOVA) was employed for statistical analysis, and Duncan’s multiple range test was utilized for multiple comparisons when significant differences were detected (*p* < 0.05). For data that did not meet assumptions of normality or homogeneity of variances, the Kruskal–Wallis H nonparametric test was applied, followed by the Games–Howell nonparametric multiple comparison test. All statistical analyses were performed using SPSS 16.0 software.

## 3. Results

### 3.1. Growth Performance

The growth performance of juvenile *C. carpio* is shown in Table 3. Replacing fishmeal with American cockroach residue in the diet did not affect the survival rate (SR) across any groups. In terms of growth indicators, the experimental groups showed a slightly higher condition factor (CF), visceral somatic index (VSI), and feed conversion ratio (FCR) than the control group, though none of these differences were statistically significant. Diet 3, which incorporated a 40% replacement of fishmeal with cockroach residue, yielded marginally higher values for final body weight (FBW), weight gain rate (WGR), specific growth rate (SGR), visceral somatic index (VSI), hepatosomatic index (HSI), and protein efficiency ratio (PER) compared to the other treatment groups; however, these differences were not statistically significant.

### 3.2. Proximate Composition and Amino Acid Content of Whole Fish

The proximate composition and amino acid profile of whole juvenile *C. carpio* are presented in Table 4. No significant differences were observed in crude protein or ash content among the five dietary groups. The moisture content of Diet 5 was significantly lower than that of Diet 2, while the crude fat content of Diet 2 was significantly lower than that of Diet 5. Regarding amino acid composition, Diet 5 exhibited significantly higher histidine content compared to the other four groups, while Diet 2 showed the highest phenylalanine content. When the fishmeal replacement level exceeded 20%, the cysteine content in the fish body showed an upward trend, with Diet 5 exhibiting significantly higher cysteine content than Diet 2. No significant differences were observed in most other amino acids, total essential amino acids (∑EAAs), total non-essential amino acids (∑NEAAs), total amino acids (∑AAs), and the ∑EAAs/∑AAs ratio.

### 3.3. Digestive Enzyme Activity in the Liver and Intestine

The activity of digestive enzymes in the liver and intestine of juvenile *C. carpio* is presented in Figure 1. In terms of liver digestive enzyme activity, no significant differences were observed in lipase or α-amylase activities among the dietary groups. However, Diet 2 exhibited the highest trypsin activity. Regarding intestinal digestive enzyme activity, a significant decreasing trend was noted in intestinal trypsin activity as the fishmeal replacement ratio increased. Additionally, Diet 2 had the highest intestinal lipase activity. The α-amylase activity in the intestines of the Diet 3 and Diet 4 groups was significantly lower than that in the control group.

### 3.4. Metabolic Parameters in the Liver and Serum

Table 5 presents the metabolic parameters in the liver and serum of juvenile *C. carpio*. Regarding serum metabolic parameters, Diet 5 exhibited significantly higher serum triglyceride (TG) content compared to Diet 2 (*p* < 0.05). However, no significant differences were observed in serum total dissoluble protein (TP), total amino acids (TAAs), urea nitrogen, total cholesterol (TC), glucose content, or the activity of alanine aminotransferase (ALT) and aspartate aminotransferase (AST) among the five dietary groups. In terms of liver metabolic parameters, the Diet 3 and Diet 5 groups showed significantly lower liver AST activity compared to Diet 1 (*p* < 0.05), while liver glucose content exhibited an increasing trend with higher levels of fishmeal replacement. No significant differences were found in liver ALT activity, TAAs, blood urea nitrogen (BUN), TG, or TC content among the five dietary groups.

### 3.5. Midgut Histology

Table 6 and Figure 2 present the effects of replacing fishmeal with American cockroach residue on the intestinal morphology of juvenile *C. carpio*. The number of goblet cells decreased as the proportion of American cockroach residue in the diet increased, with Diet 4 showing significantly lower goblet cell counts compared to the control group. However, no significant differences were observed in the number of villi, muscle layer thickness, villus height, villus width, or crypt depth among the dietary groups.

### 3.6. Antioxidant Capacity and Immunity

Table 7 presents the liver and serum antioxidant capacity, as well as immune-related indicators, in juvenile *C. carpio*. The total superoxide dismutase (T-SOD) activity in the liver of the control group was significantly lower than that in the Diet 3, Diet 4, and Diet 5 groups (*p* < 0.05). However, no significant differences were observed in liver total antioxidant capacity (T-AOC), alkaline phosphatase (ALP) activity, or malondialdehyde (MDA) content among the five dietary groups. Serum ALP activity in the Diet 1 group was significantly higher than that in the Diet 3, Diet 4, and Diet 5 groups (*p* < 0.05). No significant differences were found in serum T-AOC, T-SOD, MDA, acid phosphatase (ACP) activity, or immunoglobulin M (IgM) levels across the five dietary groups. 

## 4. Discussions

In recent years, the increasing scarcity and rising cost of fishmeal have highlighted the need for alternative high-quality protein sources in aquaculture. Numerous studies have demonstrated that various insect meals including mealworm (*Tenbrio molitor*) larvae meal, black soldier fly (*Hermetia illucens*) meal [32], housefly (*Musca domestica,* L.) [33], silkworm (*Bombyx mori*) meal [12], trogossitidae meal, grubworm meal [34], and cricket (*Acheta domesticus*) meal [35], can partially or completely replace fishmeal. Previous research has shown that substituting portions of fishmeal in carp diets with defatted black soldier fly larvae meal [36], silkworm pupae [30], and mealworm meal [12] does not negatively impact growth performance [37], suggesting that carp effectively utilize insect-based diets. Long et al. (2024) found that replacing fishmeal with American cockroach residue did not adversely affect the growth performance or body composition of juvenile Nile tilapia [14]. Additionally, Jabir et al. (2012) observed that tilapia fed diets where one-quarter of the fishmeal was substituted with mealworm meal had significantly enhanced growth and body composition [38]. Rahman et al. (1996) reported that partial or full replacement of fishmeal with silkworm pupae meal in carp diets did not harm growth performance or feed utilization efficiency [39]. 

In this study, various levels of American cockroach residue were tested as fishmeal substitutes in juvenile *C. carpio* diets, with no adverse effects on growth performance observed. With increasing levels of cockroach residue, final body weight, weight gain rate, specific growth rate, and protein efficiency ratio initially decreased, then increased, reaching a peak at 40% replacement. These findings suggest that moderate levels of cockroach residue can effectively replace fishmeal; however, excessive replacement may reduce feed utilization efficiency in juvenile *C. carpio*, likely due to anti-nutritional factors (e.g., chitin) present in cockroach residue, which juvenile *C. carpio* may have a limited ability to digest and absorb. As the fishmeal replacement levels increase, these anti-nutritional factors also increase, leading to decreased feed efficiency.

In practical aquaculture applications, a 40% replacement level with American cockroach residue is promising not only for growth performance but also for its potential economic advantages. The high cost of fishmeal is a challenge in many aquaculture sectors, while cockroach residue, as an inexpensive protein source, offers potential economic benefits. Compared to other insect protein sources, such as mealworm and black soldier fly, American cockroach residue provides a relatively balanced amino acid composition similar to that of fishmeal, making it nutritionally suitable for fish. Previous studies indicate that moderate fishmeal replacement can support adequate nutrient intake and even improve fish health by enhancing antioxidant capacity and immune function. Thus, from both an economic and nutritional perspective, American cockroach residue holds substantial potential as a fishmeal substitute. However, due to the presence of anti-nutritional factors in cockroach residue, replacement levels above 40% may adversely affect feed efficiency. Consequently, in practical production settings, maintaining a 20–40% replacement level is recommended to optimize the balance between nutritional quality, cost-effectiveness, and growth performance.

Moreover, past studies have demonstrated that excessive levels of black soldier fly and mealworm meals can impair feed efficiency in species such as rainbow trout (*Oncorhynchus mykiss*), channel catfish (*Ietalurus punetaus*), European turbot (*Scophthalmus maximus*), gilt-head bream (*Sparus aurata*), and European sea bass (*Dicentrarchus labrax*) [40,41,42,43,44]. This reduction in feed efficiency is likely due to the oxidation of insect meal fat, high chitin content, and the presence of essential oils, flavonoids, and terpenoids [5,43,45]. These factors highlight the need to carefully control replacement levels to avoid the potential negative impacts of anti-nutritional components in insect meals, thus maintaining optimal health and growth in aquaculture species. 

With the continuous expansion of aquaculture, commercially available feeds have increasingly met the nutritional and energy requirements of aquatic animals. Generally, the proximate composition of whole fish reflects the nutrient content of aquatic feed [46,47]. In this study, substituting fishmeal with American cockroach residue did not significantly alter the crude protein or ash content in the whole body of juvenile *C. carpio*; however, the crude fat content in Diet 5 was significantly greater than that in Diet 2. Regarding amino acid composition, most amino acids, total essential amino acids (∑EAAs), total non-essential amino acids (∑NEAAs), and total amino acids (∑AAs) content exhibited no significant differences among the groups. Notably, Nandeesha et al. (2000) found that replacing fishmeal with silkworm pupae meal in carp increased the crude protein and ash content significantly while decreasing crude fat content [48]. The results of this study contradict these findings, potentially due to differences in fish size, feed composition and nutrient content, insect species, and experimental conditions. In summary, substituting fishmeal with American cockroach residue did not adversely impact amino acid deposition in juvenile *C. carpio*, likely because American cockroach residue possesses protein content and amino acid composition comparable to fishmeal, facilitating nutrient digestion and absorption.

Digestive enzyme activity in fish directly influences nutrient digestion and absorption, thus affecting growth and development [49]. Trypsin, lipase, and α-amylase are key components of fish digestive enzymes [50]. Belghit et al. (2019) reported that replacing fishmeal with black soldier fly larvae meal did not significantly alter intestinal trypsin activity in Atlantic salmon [17]. The current study demonstrated that intestinal trypsin, lipase, and α-amylase activity, along with liver trypsin activity, were greater in the Diet 1 and Diet 2 groups of juvenile *C. carpio*. However, when fishmeal replacement exceeded 20%, intestinal trypsin activity was significantly lower than that in the Diet 1 group. At a 60% fishmeal replacement level, intestinal lipase and α-amylase activity were significantly lower than in the control group, but liver lipase and α-amylase activity were higher. The decrease in intestinal lipase and α-amylase activity may be due to the increased levels of anti-nutritional factors, such as chitin, in the American cockroach meal as its inclusion level rises, which could interfere with the optimal function of digestive enzymes. Overall, when the fishmeal replacement level exceeds 20%, it may negatively influence protein digestion in juvenile *C. carpio*, with a 60% replacement possibly affecting lipid and starch digestion in the intestines, although it does not significantly impact lipid and starch digestion in the liver.

Serum biochemical indices serve as critical indicators for assessing fish health [51]. Aspartate aminotransferase (AST) and alanine aminotransferase (ALT) are vital transaminases in aquatic animals, reflecting the extent of amino acid metabolism and liver damage [52]. These transaminases predominantly reside in the liver; therefore, increased liver cell is damaged, and cell membrane permeability due to damage leads to elevated transaminase activity in the blood [53]. This study found no significant differences in liver ALT activity among dietary groups; however, liver aspartate aminotransferase activity in the Diet 3 and Diet 5 groups was significantly lower than in the control group. The decrease in liver AST activity may be related to certain components in the American cockroach meal. One possible explanation is the presence of anti-nutritional factors, such as chitin, which could impact liver function and metabolic processes. Chitin, a major component of the cockroach exoskeleton, is known to have various effects on the digestive and metabolic systems of animals. It may interfere with nutrient absorption or trigger an inflammatory response, which could lead to a reduction in AST activity, as the liver is involved in detoxifying and processing such compounds. Additionally, other bioactive compounds in the cockroach meal, including certain proteins or lipids, may also affect liver enzyme activity. These components could exert a stress response on the liver, influencing enzyme production or altering the enzymatic pathways involved in liver function [54,55], indicating a potential damaging effect on fish liver at 40% and 80% fishmeal replacement levels. Urea nitrogen serves as the main metabolic product of protein metabolism, reflecting the protein metabolism status in the body, and is mainly found in the liver [56]. Although ammonia is the primary product of protein metabolism in fishes, urea, as a metabolic product of ammonia, is also present in the serum to some extent. In this study, we chose to measure serum urea instead of serum ammonia for several reasons. First, urea is more stable and easier to measure than ammonia. Ammonia is volatile in aqueous environments and has high toxicity, which makes it prone to interference during measurement. In contrast, urea remains stable in the bloodstream and can more reliably reflect protein metabolism and renal function in fish [57,58]. Additionally, serum urea levels are less affected by external factors, such as environmental conditions, which could complicate ammonia measurements. Total dissoluble protein (TP) content indicates the animal protein metabolic state and serves as a key nutritional status indicator [59]. There were no significant differences in serum BUN and TP levels among the groups, suggesting that dietary fishmeal replacement with American cockroach residue has no adverse effects on protein metabolism in *C. carpio*. Triglycerides (TGs) constitute a major component of blood lipids and are crucial indicators of fat metabolism and overall health [29,60]. This study revealed that serum TG content significantly increased when the American cockroach residue replaced fishmeal at an 80% level, indicating that excessive insect meal diminishes lipid absorption and utilization. In conclusion, when fishmeal replacement exceeds 40%, protein synthesis in fish is compromised, promoting catabolism, and levels of 40% to 80% replacement could harm liver function.

The intestine serves as the primary organ for digestion and absorption in fish, with intestinal epithelial cells and histological structure playing vital roles in nutrient assimilation [61]. Intestinal villus height, crypt depth, and muscle thickness serve as important for intestinal digestive and absorptive capacity [62]. Previous studies have shown that replacing fishmeal with cottonseed protein significantly reduces villus height, width, and muscle thickness in grouper (*Epinephelus lanceolatus*) when the replacement ratio exceeds 36% [63]. Liu et al. (2021) found that replacing 75% of fishmeal with cottonseed protein significantly reduced villus height and width in largemouth bass (*Micropterus salmoides*) [64]. Li et al. (2017) indicated that replacing fishmeal with black soldier fly pupae at levels above 75% led to severe microvilli damage in the intestines of *C. carpio*, evidenced by fragmentation and breakage [65]. The current study found that substituting fishmeal with American cockroach residue did not significantly detract from intestinal villus height, muscle thickness, or crypt depth in juvenile *C. carpio*, differing from previous findings in other fish species. This discrepancy may stem from differences in the experimental fish type and source of fishmeal replacement, leading to variations in digestion, absorption, and nutrient utilization. Goblet cells in the intestine, differentiated columnar epithelial cells, produce mucus that prevents dehydration, lubricates, and protects the gastrointestinal tract [66,67]. Mucus, primarily mucin, plays a crucial role in innate immune defense mechanisms and gut health [68]. Firstly, the mucus layer acts as a physical barrier, preventing pathogens, toxins, and other harmful substances from directly interacting with epithelial cells, thereby contributing to mucosal immunity. Secondly, the mucins secreted by goblet cells can bind to and neutralize harmful microorganisms, facilitating their clearance. Additionally, the mucus layer helps maintain the balance of the gut microbiota, which is crucial for optimizing immune function and nutrient metabolism [69,70]. Previous research observed that adding black soldier fly larvae meal increased goblet cells numbers in fish intestines [51]. Other studies demonstrated that oxidative damage resulted in decreased goblet cells counts in grouper intestines [47]. The current findings indicated that at a 60% fishmeal replacement level, the number of goblet cells in juvenile *C. carpio*’s intestines was significantly lower than that in the control group, suggesting that high fishmeal replacement levels may compromise intestinal barrier function; this could lead to increased intestinal permeability, heightened susceptibility to infections, and potentially reduced efficiency in nutrient absorption. Consequently, when replacing fishmeal with American cockroach residue, careful consideration of the replacement level is crucial.

During normal metabolic processes, the dynamic balance of reactive oxygen species (ROS) production and elimination is maintained in fish [71]. Generally, a fish’s antioxidant defense system mitigates ROS production through antioxidant enzymes [72]. Total antioxidant capacity (T-AOC) serves as a comprehensive indicator of fish antioxidant capability, while malondialdehyde (MDA), a metabolic product of cellular lipid oxidation, reflects lipid peroxidation levels [73,74]. Total superoxide dismutase (T-SOD) is a vital antioxidant enzyme that scavenges excess free radicals, protecting cells from damage [75]. This study demonstrated that at a 40% fishmeal replacement level, serum and liver T-SOD activity in juvenile *C. carpio* was elevated, while liver T-AOC activity peaked at a 20% fishmeal replacement level; liver MDA content was lowest in the Diet 2 and Diet 5 groups. These findings indicate that American cockroach residue substitution can enhance the antioxidant capacity of juvenile *C. carpio*, potentially due to the presence of active substances such as antimicrobial peptides and chitosan. 

Alkaline phosphatase (ALP), a hydrolase, disrupts pathogen structure and enhances immunity [76]. Immunoglobulin M (IgM), produced by B lymphocytes during plasma cell differentiation following antigen stimulation, signifies immune responses [77]. In this study, serum ALP activity in the Diet 2 group was not significantly different from the control group; however, when fishmeal replacement surpassed 20%, serum ALP activity decreased significantly compared to the control group. IgM content exhibited an increasing trend with rising fishmeal replacement levels, suggesting that low levels of American cockroach residue can enhance juvenile *C. carpio*’s immunity. In summary, the results illustrate that replacing 20–40% of fishmeal with American cockroach residue can improve the antioxidant capacity and immunity of juvenile *C. carpio*.

## 5. Conclusions

In conclusion, American cockroach residue can effectively substitute for fishmeal without negatively impacting the growth performance or body composition of juvenile *C. carpio* within a specific limit. A replacement level of 20% to 40% can enhance the antioxidant capacity and immunity of juvenile *C. carpio*. However, exceeding 40% replacement may harm the intestinal health of *C. carpio*, evidenced by a reduction in goblet cell numbers, which adversely affects intestinal barrier function. Therefore, it is recommended to control the replacement level of American cockroach residue in practical applications to optimize aquaculture outcomes.

## Figures and Tables

**Figure 1 animals-14-03632-f001:**
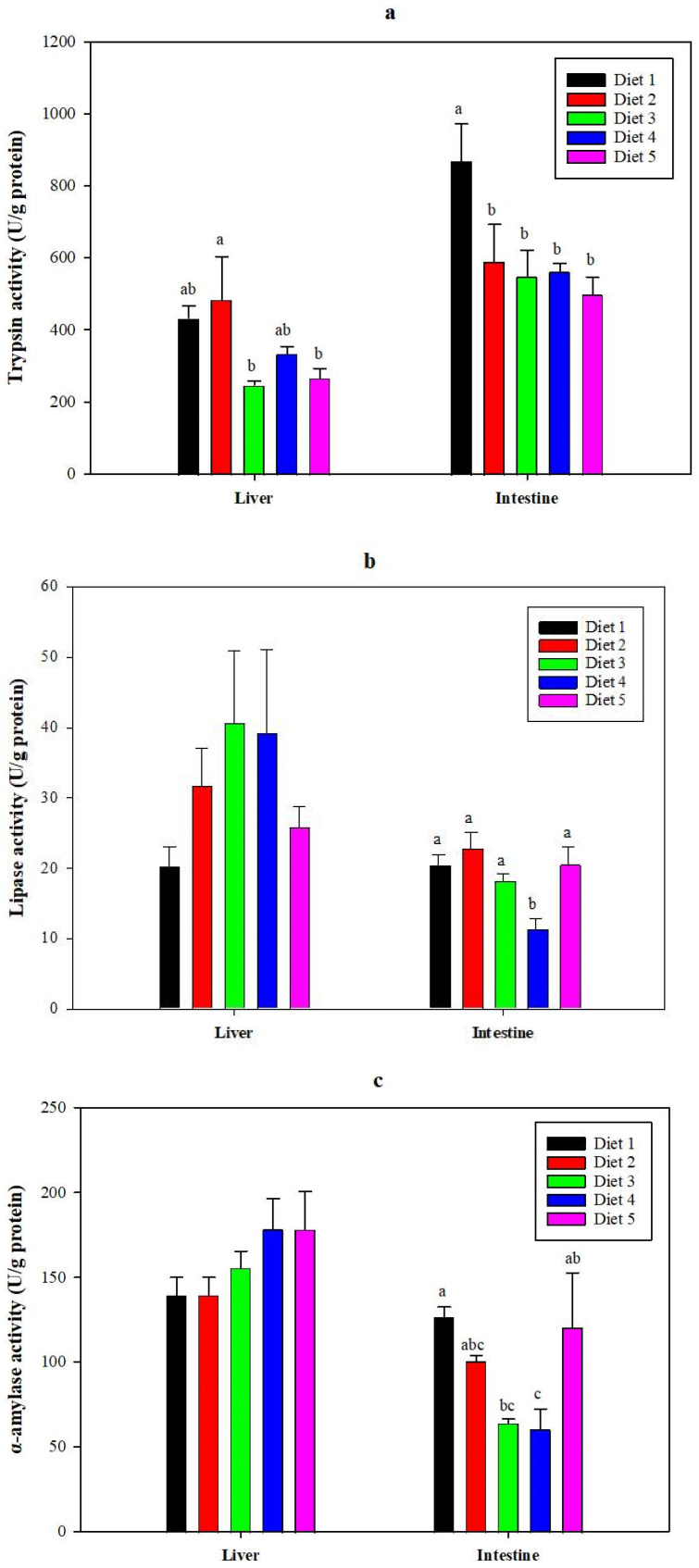
The activities of trypsin (**a**), lipase (**b**), and α-amylase (**c**) in the liver and intestine of juvenile *Cyprinus carpio.* Data are presented as mean ± SE (*n* = 3). The bars with different letters are significantly different according to Duncan’s test (*p* < 0.05), while bars without letters or with the same letters are not significantly different (*p* > 0.05).

**Figure 2 animals-14-03632-f002:**
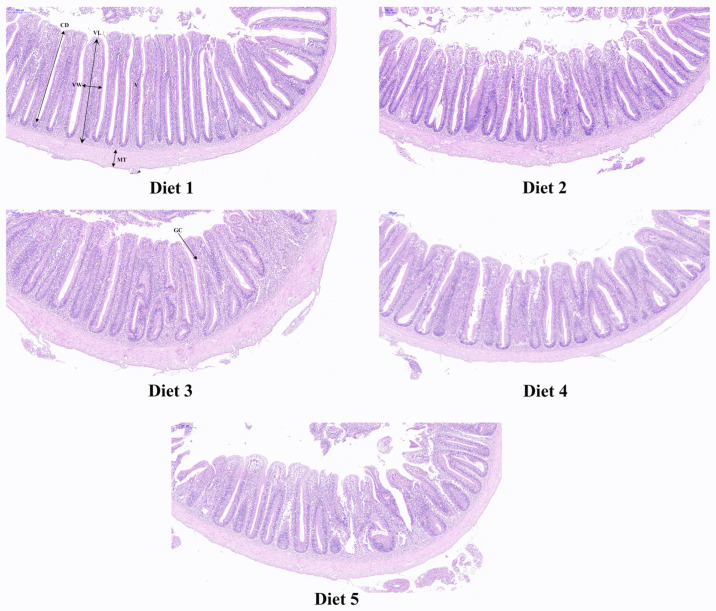
The intestinal morphology of juvenile *Cyprinus carpio*. GC: goblet cell; MT: muscle thickness; V: villi; VL: villus length; VW: villus width; CD: crypt depth.

**Table 1 animals-14-03632-t001:** The formulations of experimental diets.

Ingredients (%)	Diet 1	Diet 2	Diet 3	Diet 4	Diet 5
Fishmeal	15	12	9	6	3
*Periplaneta americana* residue	0	3	6	9	12
Soybean meal	27	27	27	27	27
Wheat flour	25	24.9	24.8	24.7	24.6
Corn gluten meal	5	5.65	6.35	7	7.7
Rapeseed meal	15	15	15	15	15
Peanut meal	5.65	5.65	5.65	5.65	5.65
Fish oil	2	1.45	0.85	0.3	0
Soybean oil	1	1	1	1	0.7
Soy lecithin	0.5	0.5	0.5	0.5	0.5
Vitamin premix ^1^	1	1	1	1	1
Mineral premix ^2^	1	1	1	1	1
Vitamin C phosphate (30%)	0.05	0.05	0.05	0.05	0.05
Choline chloride (50%)	0.25	0.25	0.25	0.25	0.25
Ethoxyquinoline	0.05	0.05	0.05	0.05	0.05
Ca(H_2_PO_4_)_2_	1.5	1.5	1.5	1.5	1.5

^1^ Vitamin premix (per kg diet): vitamin A, 8000 IU; vitamin D_3_, 4400 IU; vitamin E, 50 mg; vitamin K_3_, 6 mg; vitamin B_1_, 14 mg; vitamin B_2_, 16 mg; vitamin B_6_, 16 mg; vitamin B_12_, 0.05 mg; biotin, 0.1 mg; D-calcium pantothenate, 30 mg; folic acid, 3 mg; nicotinamide, 40 mg; vitamin C, 310 mg. ^2^ Mineral premix (per kg diet): FeSO_4_·H_2_O, 280 mg; CuSO_4_·5H_2_O, 40 mg; ZnSO_4_·H_2_O, 60 mg; MnSO_4_·H_2_O, 60 mg; MgSO_4_·H_2_O, 150 mg; KH_2_PO_4_, 2 g; NaH_2_PO_4_, 0.2 g; KI, 5 mg; CoCl_2_·6H_2_O, 5 mg; Na_2_SeO_3_, 4 mg.

**Table 2 animals-14-03632-t002:** Proximate compositions and amino acids content of experimental diets.

Items (% Dry Diet)	Diet 1	Diet 2	Diet 3	Diet 4	Diet 5
Proximate compositions (% dry diet)					
Dry matter	88.05	88.07	88.31	87.04	87.42
Crude protein	33.65	33.59	33.67	33.81	33.88
Crude fat	5.47	5.36	5.44	5.48	5.43
Ash	12.00	10.30	9.20	10.10	8.50
Amino acids contents (% dry diet)					
Arginine	2.03	1.95	2.03	1.91	1.96
Histidine	0.54	0.57	0.61	0.63	0.68
Isoleucine	1.54	1.51	1.57	1.54	1.59
Leucine	2.58	2.58	2.69	2.67	2.78
Lysine	1.48	1.45	1.48	1.45	1.51
Methionine	0.36	0.31	0.37	0.34	0.32
Phenylalanine	1.42	1.47	1.56	1.61	1.71
Threonine	1.21	1.21	1.24	1.22	1.27
Tryptophan	0.40	0.39	0.41	0.40	0.41
Valine	1.81	1.79	1.87	1.85	1.88
∑EAAs	13.37	13.23	13.83	13.63	14.11
Alanine	1.70	1.74	1.85	1.90	1.99
Aspartic acid	2.59	2.57	2.69	2.65	2.81
Cysteine	0.32	0.28	0.31	0.28	0.27
Glutamic acid	5.66	5.63	5.99	5.94	6.16
Glycine	1.64	1.57	1.54	1.45	1.41
Proline	5.95	5.66	5.78	5.53	5.82
Serine	1.94	1.84	1.81	1.70	1.68
Tyrosine	0.92	0.95	1.03	1.05	1.13
∑NEAAs	20.73	20.25	21.01	20.50	21.28
∑AAs	34.10	33.47	34.85	34.12	35.38
∑EAAs/∑AAs	0.39	0.40	0.40	0.40	0.40

∑EAA: total essential amino acids; ∑NEAA: total non-essential amino acids; ∑AA: total amino acids.

**Table 3 animals-14-03632-t003:** Growth performance of juvenile *Cyprinus carpio*.

Items	Diet 1	Diet 2	Diet 3	Diet 4	Diet 5
SR (%)	93.33 ± 4.41	95.00 ± 2.89	86.67 ± 13.33	85.00 ± 12.5	90.00 ± 7.64
IBW (g)	74.13 ± 0.03	74.24 ± 0.04	74.15 ± 0.07	73.82 ± 0.38	74.19 ± 0.05
FBW (g)	105.41 ± 9.38	102.08 ± 1.09	109.67 ± 4.49	105.89 ± 11.53	100.44 ± 4.66
WGR (%)	42.19 ± 12.69	37.59 ± 1.45	47.87 ± 6.11	43.37 ± 15.20	35.37 ± 6.27
SGR (%/d)	0.56 ± 0.14	0.52 ± 0.02	0.64 ± 0.07	0.57 ± 0.17	0.49 ± 0.08
FCR (%)	1.77 ± 0.39	1.89 ± 0.21	1.87 ± 0.42	1.98 ± 0.44	1.80 ± 0.07
PER	2.47 ± 0.74	2.06 ± 0.08	2.55 ± 0.32	2.21 ± 0.78	1.93 ± 0.34
CF (g/cm^3^%)	2.20 ± 0.04	2.38 ± 0.08	2.27 ± 0.09	2.32 ± 0.10	2.35 ± 0.05
VSI (%)	11.13 ± 0.93	11.76 ± 0.56	12.22 ± 0.23	11.83 ± 1.26	11.68 ± 0.17
HSI (%)	3.20 ± 0.11	3.42 ± 0.32	3.63 ± 0.17	3.06 ± 0.14	3.15 ± 0.17

Data are presented as mean ± SE (*n* = 3). Values within the same row with different letters indicate a significant difference (*p* < 0.05), while values within the same row without letters or with the same letters indicate no significant difference (*p* > 0.05). SR: survival rate; IBW: initial body weight; FBW: final body weight; WGR: weight gain rate; SGR: specific growth rate; FCR: feed conversion ratio; PER: protein efficiency ratio; CF: condition factor; VSI: viscerosomatic index; HSI: hepatosomatic index.

**Table 4 animals-14-03632-t004:** Proximate compositions and amino acids contents of whole body of juvenile *Cyprinus carpio*.

Items	Diet 1	Diet 2	Diet 3	Diet 4	Diet 5
Proximate compositions (% wet matter)					
Moisture	79.36 ± 0.80 ^ab^	80.97 ± 1.24 ^a^	78.71 ± 1.83 ^ab^	77.93 ± 1.52 ^ab^	76.31 ± 0.64 ^b^
Crude protein	15.63 ± 0.09	15.60 ± 0.55	15.52 ± 0.37	15.57 ± 0.81	16.22 ± 0.32
Crude fat	6.05 ± 0.93 ^ab^	5.35 ± 0.64 ^b^	7.21 ± 0.69 ^ab^	7.05 ± 0.56 ^ab^	7.92 ± 0.42 ^a^
Ash	3.26 ± 0.26	3.57 ± 0.12	3.22 ± 0.06	3.45 ± 0.29	3.09 ± 1.08
Amino acids contents (% dry matter)					
Arginine	3.33 ± 0.10	3.48 ± 0.24	3.83 ± 0.34	3.23 ± 0.14	3.71 ± 0.03
Histidine	1.07 ± 0.06 ^b^	1.11 ± 0.02 ^b^	1.15 ± 0.06 ^b^	1.12 ± 0.02 ^b^	1.43 ± 0.15 ^a^
Isoleucine	2.56 ± 0.05	2.71 ± 0.10	2.71 ± 0.14	2.57 ± 0.06	2.81 ± 0.10
Leucine	4.37 ± 0.04	4.56 ± 0.16	4.58 ± 0.16	4.41 ± 0.10	4.58 ± 0.04
Lysine	4.89 ± 0.08	5.13 ± 0.16	5.15 ± 0.19	5.20 ± 0.20	5.36 ± 0.11
Methionine	1.04 ± 0.09	1.21 ± 0.04	1.21 ± 0.08	1.26 ± 0.04	1.38 ± 0.21
Phenylalanine	2.08 ± 0.02 ^b^	2.32 ± 0.11 ^a^	2.25 ± 0.07 ^ab^	2.12 ± 0.06 ^ab^	2.10 ± 0.05 ^ab^
Threonine	2.34 ± 0.04	2.44 ± 0.08	2.44 ± 0.09	2.33 ± 0.05	2.32 ± 0.04
Tryptophan	0.71 ± 0.01	0.75 ± 0.03	0.75 ± 0.03	0.72 ± 0.03	0.74 ± 0.01
Valine	3.29 ± 0.02	3.44 ± 0.08	3.48 ± 0.10	3.36 ± 0.11	3.56 ± 0.20
∑EAAs	25.68 ± 0.41	27.15 ± 0.97	27.54 ± 1.05	26.30 ± 0.68	28.00 ± 0.64
Alanine	4.35 ± 0.05	4.53 ± 0.20	4.48 ± 0.14	4.40 ± 0.21	4.48 ± 0.12
Aspartic acid	5.63 ± 0.06	5.90 ± 0.21	5.88 ± 0.18	5.62 ± 0.13	5.84 ± 0.13
Cysteine	0.45 ± 0.02 ^ab^	0.40 ± 0.01 ^b^	0.52 ± 0.04 ^ab^	0.56 ± 0.07 ^ab^	0.64 ± 0.12 ^a^
Glutamic acid	8.44 ± 0.11	8.94 ± 0.34	8.87 ± 0.33	8.53 ± 0.28	8.76 ± 0.14
Glycine	4.10 ± 0.09	4.44 ± 0.24	4.38 ± 0.20	4.22 ± 0.25	4.08 ± 0.06
Proline	7.62 ± 0.33	8.22 ± 0.64	8.27 ± 0.39	7.33 ± 0.52	6.99 ± 0.39
Serine	2.85 ± 0.03	2.93 ± 0.09	2.99 ± 0.13	2.85 ± 0.10	2.75 ± 0.04
Tyrosine	1.66 ± 0.01	1.77 ± 0.06	1.78 ± 0.06	1.64 ± 0.04	1.88 ± 0.13
∑NEAAs	35.10 ± 0.59	37.13 ± 1.72	37.17 ± 1.38	35.16 ± 1.56	35.41 ± 0.31
∑AAs	60.78 ± 0.87	64.28 ± 2.68	64.72 ± 2.42	61.46 ± 2.22	63.41 ± 0.72
∑EAAs/∑AAs	0.42 ± 0.01	0.42 ± 0.01	0.43 ± 0.01	0.43 ± 0.01	0.44 ± 0.01

Data are presented as mean ± SE (*n* = 3). Values within the same row with different letters indicate a significant difference (*p* < 0.05), while values within the same row without letters or with the same letters indicates no significant difference (*p* > 0.05). ∑EAA: total essential amino acids; ∑NEAAs: total non-essential amino acids; ∑AAs: total amino acids.

**Table 5 animals-14-03632-t005:** Metabolic indices in the liver and intestine of juvenile *Cyprinus carpio*.

Items	Diet 1	Diet 2	Diet 3	Diet 4	Diet 5
Serum					
TP (g/L)	21.62 ± 4.59	21.73 ± 1.39	23.20 ± 2.61	25.88 ± 4.34	31.09 ± 2.51
ALT (U/L)	9.66 ± 0.48	10.75 ± 0.53	11.99 ± 1.63	13.75 ± 2.80	10.95 ± 0.38
AST (U/L)	16.75 ± 1.67	34.66 ± 10.88	31.81 ± 10.86	23.24 ± 4.11	21.56 ± 3.31
TAAs (mmol/L)	29.86 ± 1.84	29.69 ± 2.41	31.94 ± 1.66	34.38 ± 4.21	32.12 ± 0.87
BUN (mmol/L)	2.43 ± 0.26	2.78 ± 0.41	3.42 ± 0.31	2.98 ± 0.23	3.28 ± 0.19
TC (mmol/L)	4.46 ± 0.55	4.07 ± 0.52	4.39 ± 0.48	3.45 ± 0.48	4.08 ± 0.23
TG (mmol/L)	2.21 ± 0.40 ^ab^	1.85 ± 0.16 ^b^	2.27 ± 0.17 ^ab^	2.07 ± 0.34 ^ab^	2.87 ± 0.07 ^a^
GLU (mmol/L)	8.66 ± 1.77	9.20 ± 0.71	9.89 ± 1.32	9.40 ± 2.04	6.74 ± 0.49
Liver					
ALT (U/g protein)	8.45 ± 2.88	12.40 ± 1.15	13.44 ± 3.56	18.37 ± 4.79	12.56 ± 1.16
AST (U/g protein)	17.98 ± 1.30 ^a^	16.10 ± 2.01 ^ab^	12.34 ± 0.97 ^bc^	18.70 ± 0.14 ^a^	8.27 ± 0.63 ^c^
BUN (mmol/g protein)	0.47 ± 0.03	0.51 ± 0.02	0.53 ± 0.12	0.61 ± 0.05	0.57 ± 0.01
T-AAs (mmol/g protein)	12.79 ± 3.22	13.14 ± 0.62	20.60 ± 6.26	24.30 ± 7.82	14.54 ± 4.49
TG (mmol/kg tissue)	7.95 ± 2.58	6.49 ± 0.57	10.07 ± 2.27	8.34 ± 0.18	7.74 ± 1.40
TC (mmol/kg tissue)	5.48 ± 0.71	5.37 ± 0.44	5.60 ± 0.81	4.92 ± 0.24	4.53 ± 0.26
GLU (mmol/g protein)	0.39 ± 0.08 ^b^	0.49 ± 0.02 ^ab^	0.66 ± 0.09 ^a^	0.74 ± 0.11 ^a^	0.68 ± 0.05 ^a^

Data are presented as mean ± SE (*n* = 3). Values within the same row with different letters indicate a significant difference (*p* < 0.05), while values within the same row without letters or with the same letters indicate no significant difference (*p* > 0.05). TP: total dissoluble protein; ALT: alanine aminotransferase; AST: aspartate aminotransferase; T-AAs: total amino acids; BUN: blood urea nitrogen; TC: total cholesterol; TG: triglyceride; GLU: glucose.

**Table 6 animals-14-03632-t006:** Midgut histology of juvenile *Cyprinus carpio*.

Items	Diet 1	Diet 2	Diet 3	Diet 4	Diet 5
Villus number	37.83 ± 3.78	32.50 ± 3.88	30.83 ± 3.31	32.58 ± 4.78	28.83 ± 5.46
Muscle thickness (μm)	69.78 ± 3.61	69.84 ± 6.26	65.63 ± 4.99	68.25 ± 4.83	69.43 ± 4.93
Villus length (μm)	654.71 ± 79.04	635.64 ± 42.58	617.70 ± 50.47	636.67 ± 47.61	663.65 ± 35.88
Villus width (μm)	122.92 ± 4.31	120.53 ± 9.92	131.36 ± 4.17	119.83 ± 3.03	134.04 ± 7.03
Crypt depth (μm)	442.46 ± 40.90	439.93 ± 52.34	405.32 ± 30.89	445.86 ± 40.53	495.25 ± 36.03
Goblet cell number	26.65 ± 3.52 ^a^	22.57 ± 3.73 ^ab^	19.75 ± 2.91 ^ab^	13.88 ± 1.52 ^b^	16.19 ± 1.55 ^ab^

Data are presented as mean ± SE (*n* = 3). Values within the same row with different letters indicate a significant difference (*p* < 0.05), while values within the same row without letters or with the same letters indicate no significant difference (*p* > 0.05).

**Table 7 animals-14-03632-t007:** Metabolic indices in the liver and intestine of juvenile *Cyprinus carpio*.

Items	Diet 1	Diet 2	Diet 3	Diet 4	Diet 5
Serum					
T-AOC (U/L)	4.23 ± 0.31	4.05 ± 0.34	4.17 ± 0.23	4.75 ± 0.39	4.32 ± 0.26
T-SOD (U/mL)	266.97 ± 48.54	245.72 ± 33.02	297.97 ± 20.95	277.87 ± 13.13	298.54 ± 41.65
MDA (nmol/mL)	8.08 ± 1.22	7.71 ± 1.01	8.83 ± 0.76	8.48 ± 1.35	9.09 ± 0.88
ALP (U/100 mL)	23.09 ± 3.29 ^a^	14.89 ± 5.29 ^ab^	12.16 ± 0.76 ^b^	8.19 ± 1.33 ^b^	8.97 ± 1.33 ^b^
ACP (U/100 mL)	835.18 ± 214.36	934.73 ± 226.43	1042.43 ± 67.14	1045.93 ± 103.32	921.63 ± 123.65
IgM(μg/mL)	148.73 ± 24.79	160.31 ± 14.22	197.29 ± 19.48	200.50 ± 77.46	217.94 ± 25.95
Liver					
T-AOC (U/mg protein)	0.21 ± 0.02	0.37 ± 0.10	0.25 ± 0.03	0.31 ± 0.05	0.28 ± 0.05
T-SOD (U/mg protein)	0.83 ± 0.15 ^b^	1.11 ± 0.10 ^ab^	1.34 ± 0.02 ^a^	1.29 ± 0.18 ^a^	1.27 ± 0.09 ^a^
MDA (nmol/mg protein)	0.36 ± 0.04	0.30 ± 0.04	0.37 ± 0.16	0.48 ± 0.09	0.25 ± 0.04
ALP (U/100 g protein)	0.63 ± 0.15	0.97 ± 0.21	0.69 ± 0.07	0.94 ± 0.11	0.72 ± 0.13
ACP (U/100 g protein)	0.89 ± 0.14 ^b^	1.71 ± 0.38 ^a^	0.92 ± 0.25 ^b^	1.14 ± 0.10 ^ab^	0.94 ± 0.17 ^b^

Data are presented as mean ± SE (*n* = 3). Values within the same row with different letters indicate a significant difference (*p* < 0.05), while values within the same row without letters or with the same letters indicate no significant difference (*p* > 0.05). T-AOC: total antioxidant capacity; T-SOD: total superoxide dismutase; MDA: malondialdehyde; ALP: alkaline phosphatase; ACP: acid phosphatase; IgM: immunoglobulin M.

## Data Availability

The original contributions presented in this study are included in the article. Further inquiries can be directed to the corresponding authors.

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
