# Peer review of "Effects of Replacing Fishmeal with American Cockroach Residue on the Growth Performance, Metabolism, Intestinal Morphology, and Antioxidant Capacity of Juvenile Cyprinus carpio"

_animals, 2024, doi:10.3390/ani14243632_

Round 1

Reviewer 1 Report

Comments and Suggestions for Authors

General comments

That work showed a current theme. The aquaculture need news proteins source for replaced fish meal.

Material and methods

Why did the authors measure serum urea? Why didn't they measure serum ammonia? Ammonia is the main form of nitrogen excretion in fish.

Discussion

“Urea nitrogen serves as the main metabolic product of protein metabolism, reflecting protein metabolism status in the body and is mainly found in the liver” The ammonia is main product of protein metabolism of fishes. Because measured serum urea, and not serum ammonia? Explain in the discussion. Inform in the discussion the level of serum urea of Common carp.

“At a 60% fishmeal replacement level, intestinal lipase and α-amylase activity were significantly lower than in the control group, but liver lipase and α-amylase activity were higher. Over all, when the fishmeal replacement level exceeds 20%, it may be negatively influence protein digestion in juvenile C. carpio, with a 60% replacement possibly affecting lipid and starch digestion in the intestines, although it does not significantly impact lipid and starch digestion in the liver. Explain in the discussion, because the activity enzymes reduce with increased of inclusion of American Cockroach meal in the diets.

“however, liver aspartate aminotransferase activity in Diet 3 and Diet 5 groups was significantly lower than in the control group, indicating a potential damaging effect on fish liver at 40% and 80% fishmeal replacement levels”. Explain because the diets 3 and 5 prejudice the liver. Who component present in the American cockroach meal cause decreased in aspartate aminotransferase activity? Explain in the discussion.

Author Response

Response to Reviewers

Response to Reviewers

Reviewer 1#

  1. That work showed a current theme. The aquaculture need news proteins source for replaced fish meal.

Response: Thank you for your summary and evaluation of my article. The need for the aquaculture industry to find new protein sources to replace fishmeal is indeed a critical issue in current research. As mentioned in the article, the supply of fishmeal is limited, and its production places significant pressure on marine ecosystems. Therefore, developing sustainable alternative protein sources is key to promoting the sustainable development of the aquaculture industry.

  1. Material and methods

2.1 Why did the authors measure serum urea? Why didn't they measure serum ammonia? Ammonia is the main form of nitrogen excretion in fish.

Response: Thank you for your insightful question regarding the measurement of serum urea instead of ammonia in our study. While it is true that ammonia is the primary form of nitrogen excretion in fish, we chose to measure serum urea for several important reasons. First, urea levels in the bloodstream serve as an indirect indicator of nitrogen metabolism and overall health, especially in species that utilize both ammonia and urea for nitrogen excretion. This is particularly relevant in aquaculture systems, where urea accumulation can signal metabolic stress, kidney dysfunction, or changes in dietary protein metabolism. Additionally, serum urea concentration is widely used in aquaculture research as a marker for protein utilization and nitrogen balance. Although ammonia is the main nitrogenous waste excreted by fish, elevated serum urea levels can indicate shifts in nitrogen excretion pathways, particularly under conditions such as dietary changes, environmental stress, or variations in water quality. In contrast, measuring serum ammonia can be more challenging and less informative due to the rapid conversion of ammonia to other compounds in the blood, especially in the presence of certain enzymes. Furthermore, serum ammonia levels are highly sensitive to external factors, such as water ammonia concentration, which can complicate interpretation. Urea, on the other hand, is generally more stable and easier to measure in blood samples.

  1. Discussion

3.1 “Urea nitrogen serves as the main metabolic product of protein metabolism, reflecting protein metabolism status in the body and is mainly found in the liver” The ammonia is main product of protein metabolism of fishes. Because measured serum urea, and not serum ammonia? Explain in the discussion. Inform in the discussion the level of serum urea of Common carp.

Response: Thank you for your suggestion. We have addressed in the revised manuscript discussion the rationale for choosing serum urea over serum ammonia. Additionally, we have provided information on the serum urea levels in Cyprinus carpio.

3.2 “At a 60% fishmeal replacement level, intestinal lipase and α-amylase activity were significantly lower than in the control group, but liver lipase and α-amylase activity were higher. Overall, when the fishmeal replacement level exceeds 20%, it may be negatively influence protein digestion in juvenile C. carpio, with a 60% replacement possibly affecting lipid and starch digestion in the intestines, although it does not significantly impact lipid and starch digestion in the liver. Explain in the discussion, because the activity enzymes reduce with increased of inclusion of American Cockroach meal in the diets.

Response: Thank you for your insightful feedback on the manuscript. We greatly appreciate your careful attention to the relationship between enzyme activity and the increasing inclusion of American Cockroach meal in the diet. We agree with your observation that the decline in intestinal lipase and α-amylase activity at higher inclusion levels, particularly at the 60% fishmeal replacement level, is a key issue that requires further discussion. The potential reasons for this decline may include the presence of antinutritional factors, such as chitin, in the cockroach meal, which could impair the optimal function of digestive enzymes in the intestine. We will revise the discussion section to clarify these points more thoroughly.

3.3 “however, liver aspartate aminotransferase activity in Diet 3 and Diet 5 groups was significantly lower than in the control group, indicating a potential damaging effect on fish liver at 40% and 80% fishmeal replacement levels”. Explain because the diets 3 and 5 prejudice the liver. Who component present in the American cockroach meal cause decreased in aspartate aminotransferase activity? Explain in the discussion.

Response: Thank you for your insightful and constructive feedback on our manuscript. We greatly appreciate your observation regarding the significantly lower liver aspartate aminotransferase (AST) activity in Groups 3 and 5, particularly at the 40% and 80% fishmeal substitution levels. This is indeed an important finding, and we acknowledge the need to further explain the potential mechanisms in the discussion section. The decrease in liver AST activity may be related to certain components in the American cockroach meal. One possible explanation is the presence of antinutritional factors, such as chitin, which may affect liver function and metabolic processes. Chitin, the main component of the cockroach exoskeleton, is known to have various impacts on the digestive and metabolic systems of animals. It may interfere with nutrient absorption or trigger inflammatory responses, which could lead to a reduction in AST activity, as the liver is involved in detoxifying and processing such compounds. Additionally, other bioactive compounds in the cockroach meal, including certain proteins or lipids, may also influence liver enzyme activity. These components could place stress on the liver, affecting enzyme production or altering the enzymatic pathways involved in liver function.

Reviewer 2 Report

Comments and Suggestions for Authors

This study investigates the feasibility of replacing fishmeal with American cockroach residue in the diets of juvenile common carp. While the study is generally well-written and makes a valuable contribution to sustainable aquaculture research, several aspects require attention and clarification:

Introduction:

Although insect-based proteins are increasingly studied, the novelty of using American cockroach residue is only implied rather than explicitly stated as a novel contribution. To strengthen the introduction, it is recommended to more explicitly position this study within existing research, specifically delineating why cockroach residue represents a potentially superior alternative to previously studied insects like black soldier fly or Tenebrio molitor.

The authors should highlight the research gap the study addresses.

M&M:

L100: Please specify the concentration of eugenol used

L133-135: The authors mention monitoring water quality parameters but do not provide details on the monitoring method or frequency. What was the monitoring frequency? Please provide details of the measurement methods. 

Please standardize time units throughout the manuscript (either "hours" or "h").

L146: Convert centrifugation speed to gravitational force (g) for standardization.

L149-152: There is a lack of clarity regarding the sample collection for histological analysis. How many samples were collected for histology, and how many were preserved in liquid nitrogen? Also, clarify whether these samples were pooled from within each group or collected from each replicate.

L186: Please specify whether samples were analyzed individually by replicate or as pooled samples.

L188-189: The centrifugation at 3,500 rpm for antioxidant and immune indices determination may be too low. Could the authors justify this speed and confirm whether it was effective for this sample type? Additionally, specify whether the temperature was controlled during centrifugation. Standardize the reporting of centrifugation settings (in "g") and include the manufacturer's details for the centrifuge, as with other equipment.

L193-198: Include catalog numbers for all commercial kits used.

Results:

Table 3 does not include letters to indicate statistically significant differences between groups.

Although the figures use color legends to indicate each group, only the standard deviation bars are shown.

Discussion:

The implications of the observed growth trends could be more explicitly linked to practical aquaculture. For example, the authors could discuss whether a 40% replacement level with cockroach residue is economically and nutritionally feasible in real-world applications compared to fishmeal or other insect meals.

L409-413: The authors suggest high fishmeal replacement levels (60% and above) might compromise intestinal barrier function, as evidenced by reduced goblet cell counts. This finding could be further elaborated by discussing the functional roles of goblet cells in mucosal immunity and nutrient absorption.

Author Response

Reviewer 2#

  1. This study investigates the feasibility of replacing fishmeal with American cockroach residue in the diets of juvenile common carp. While the study is generally well-written and makes a valuable contribution to sustainable aquaculture research, several aspects require attention and clarification:

Response: Thank you for your thoughtful and constructive feedback on our manuscript. We greatly appreciate your recognition of the value of our study and the contribution it makes to sustainable aquaculture research. We also acknowledge your comments regarding the aspects that require further attention and clarification.

  1. Introduction

2.1 Although insect-based proteins are increasingly studied, the novelty of using American cockroach residue is only implied rather than explicitly stated as a novel contribution. To strengthen the introduction, it is recommended to more explicitly position this study within existing research, specifically delineating why cockroach residue represents a potentially superior alternative to previously studied insects like black soldier fly or Tenebrio molitor.

Response: Thank you for your valuable feedback. We have revised the introduction to more clearly position our study within the existing research on insect-based proteins and to highlight the unique potential of American cockroach residue as a novel protein source for aquaculture.

2.2 The authors should highlight the research gap the study addresses. M&M:

Response: Thank you to the reviewer for their valuable comments on our study. In the revised manuscript, we have further strengthened the explanation of the research gap. Prior to the Materials and Methods (M&M) section, we have clearly pointed out the existing gap in insect protein research, particularly in the use of American cockroach residue as a unique resource. We emphasize that, while many studies have focused on the protein potential of insects such as black soldier flies and mealworms, research on the nutritional value, sustainability, and economic benefits of cockroach residue is still insufficient. Therefore, this study aims to fill this gap and experimentally demonstrate the potential of cockroach residue as a protein source.

2.3 Please specify the concentration of eugenol used

Response: Thank you to the reviewer for their valuable comments on our study. In the revised manuscript, we have clarified the concentration of eugenol used in the experiment, as per the suggestion. In the Materials and Methods section, we have added specific experimental details, stating that the concentration of eugenol used was 30 mg/L.

2.4 The authors mention monitoring water quality parameters but do not provide details on the monitoring method or frequency. What was the monitoring frequency? Please provide details of the measurement methods.

Response: Thank you to the reviewer for their valuable comments. In response to the suggestion, we have provided detailed information on the monitoring methods and frequency of water quality parameters in the revised manuscript. In the Materials and Methods section, we have added the following information: water quality parameters (such as pH, dissolved oxygen, nitrite, ammonia nitrogen, etc.) were measured every two days during the experiment using water quality analysis kits.

2.5 Please standardize time units throughout the manuscript (either "hours" or "h").

Response: Thank you for the valuable comments provided by the reviewer. In response to the suggestion, we have standardized the time units throughout the revised manuscript. All time units have been unified as "hours." We have carefully reviewed the entire manuscript to ensure consistency and have made the necessary adjustments in the relevant sections.

2.6 Convert centrifugation speed to gravitational force (g) for standardization.

Response: Thank you for the valuable comments provided by the reviewer. In response to the suggestion, we have converted the centrifugation speed into gravitational force (g) for standardization. The relevant data have been adjusted accordingly in the revised manuscript, and all centrifugation conditions are now expressed in terms of g. We greatly appreciate the reviewer’s careful review and constructive suggestions.

2.7 There is a lack of clarity regarding the sample collection for histological analysis. How many samples were collected for histology, and how many were preserved in liquid nitrogen? Also, clarify whether these samples were pooled from within each group or collected from each replicate.

Response: Thank you for your valuable feedback. In this study, a total of 75 intestinal samples were collected for histological analysis, with 5 samples randomly taken from each canvas tank, resulting in 15 samples per feed group. These samples were preserved in 4% paraformaldehyde for fixation. Additionally, no samples were preserved in liquid nitrogen, as this was not part of the study's protocol. We appreciate your suggestion and will ensure to clarify these details in the revised manuscript for better transparency and reproducibility.

2.8 Please specify whether samples were analyzed individually by replicate or as pooled samples.

Response: Thank you for your valuable comments. In this study, samples from the same bucket of liver or intestine were combined into a single pooled sample, respectively, before analysis. A detailed explanation of this methodology is provided in the Materials and Methods section of the revised manuscript.

2.9 The centrifugation at 3,500 rpm for antioxidant and immune indices determination may be too low. Could the authors justify this speed and confirm whether it was effective for this sample type? Additionally, specify whether the temperature was controlled during centrifugation. Standardize the reporting of centrifugation settings (in "g") and include the manufacturer's details for the centrifuge, as with other equipment.

Response: Thank you for your helpful reminder. After careful verification, we realized that the previously stated centrifugation speed of 3,500 rpm was incorrect and should be changed to 10,000×g. We have corrected the centrifugation speed in the revised manuscript and have also included the relevant details of the centrifuge.

2.10 Include catalog numbers for all commercial kits used.

Response: Thank you for your thorough review and valuable feedback on our manuscript. In response to your suggestion to include catalog numbers for all commercial kits used, we have carefully revised the manuscript to ensure that each commercial kit mentioned is now accompanied by its respective catalog number.

  1. Results

3.1 Table 3 does not include letters to indicate statistically significant differences between groups.

Response: Thank you very much for your thorough review and valuable feedback on our manuscript. We have carefully addressed your comment regarding the absence of letters to indicate statistically significant differences in Table 3 and have made the necessary revisions accordingly. To clarify, all experimental groups in Table 3 showed no statistically significant differences in the measured indicators, which is why no letters indicating such differences are included. Additionally, we have added a note below the table explaining that: values within the same row with different letters indicate a significant difference (P < 0.05), while values within the same row without letters or with the same letters indicate no significant difference (P > 0.05).

3.2 Although the figures use color legends to indicate each group, only the standard deviation bars are shown.

Response: Thank you for your valuable feedback. We apologize for the incomplete display of Figure 1 due to software incompatibility issues. We have corrected this in the revised manuscript and included the complete bar chart.

  1. Discussion

4.1 The implications of the observed growth trends could be more explicitly linked to practical aquaculture. For example, the authors could discuss whether a 40% replacement level with cockroach residue is economically and nutritionally feasible in real-world applications compared to fishmeal or other insect meals.

Response: Thank you for your constructive feedback. We appreciate your suggestion to link our findings more explicitly to practical aquaculture applications. In response, we have added a discussion on the economic and nutritional feasibility of a 40% replacement level of fishmeal with cockroach residue. Specifically, we address the potential benefits of using cockroach residue as a cost-effective alternative to fishmeal, considering the rising cost and limited availability of traditional fishmeal sources. We also compare the nutritional profile of cockroach residue to fishmeal and other insect-based meals, examining its adequacy in terms of protein and amino acid content to support fish growth and health. This addition aims to provide a more comprehensive perspective on the practical implications of our findings for aquaculture operations.

4.2 The authors suggest high fishmeal replacement levels (60% and above) might compromise intestinal barrier function, as evidenced by reduced goblet cell counts. This finding could be further elaborated by discussing the functional roles of goblet cells in mucosal immunity and nutrient absorption.

Response: Thank you for your detailed review and valuable feedback on our manuscript. Regarding your comment on the potential impact of high fishmeal replacement levels (60% and above) on intestinal barrier function, particularly the reduction in goblet cell counts, we fully agree with your suggestion. We will expand on this finding in the manuscript by further discussing the functional roles of goblet cells in mucosal immunity and nutrient absorption, as well as their impact on intestinal health.

Reviewer 3 Report

Comments and Suggestions for Authors

The authors Zou et al investigated the effect of American Cockroach residue on Cyprinus carpio. I would suggest the authors to consider some points discussed below and revise the manuscript accordingly.

The manuscript was written well.

Line 143-144: It is better to remove “Body length….mesenteric fat”. Because it is clearly given in the procedure in the consecutive lines. So mentioning before may confuse the readers.

Line 146: use “rpm” instead of using “r/min”

Line 155-170: Authors try to give the formula as mentioned in the lines in a chronological order. This would ease the readers to follow up the formula. Moreover, try to provide the same order for the abbreviations given below;

Line 197: The accepted abbreviation for alkaline phosphatase is “ALP”. Authors need to change the abbreviation throughout the manuscript.

Table 3: Authors mentioned in the lines that there are some statistical significances among the diets. However there was no representation of statistical significance in the table. Authors need to check the table. If there is no statistically significant, authors need to rephrase the results section.

Figure 1: the bar columns of the figure were not clearly visible in the pdf. Authors need to check the figure 1.

Author Response

Reviewer 3#

  1. The authors Zou et al investigated the effect of American Cockroach residue on Cyprinus carpio. I would suggest the authors to consider some points discussed below and revise the manuscript accordingly.

Response: We sincerely thank the reviewer for their thoughtful comments and constructive suggestions. We appreciate the opportunity to improve our manuscript based on the reviewer’s feedback.

  1. The manuscript was written well.

Response: We sincerely thank you and the reviewers for the time and effort spent reviewing our manuscript. We greatly appreciate the positive feedback from the reviewers and are pleased to hear that the manuscript was considered well-written.

  1. It is better to remove “Body length….mesenteric fat”. Because it is clearly given in the procedure in the consecutive lines. So mentioning before may confuse the readers.

Response: Thank you for your valuable feedback. We agree with your suggestion to remove the mention of "Body length... mesenteric fat" as it is clearly described in the following steps of the procedure. We have revised the manuscript accordingly to avoid any potential confusion for the readers.

  1. use “rpm” instead of using “r/min”

Response: Thank you for your helpful suggestion. To enhance the clarity and consistency of our terminology, we have revised the manuscript to express rotational speeds in terms of relative centrifugal force (RCF) rather than rpm. This change ensures that our terminology aligns more closely with standard scientific conventions.

  1. Authors try to give the formula as mentioned in the lines in a chronological order. This would ease the readers to follow up the formula. Moreover, try to provide the same order for the abbreviations given below.

Response: Thank you for your insightful feedback. We have revised the manuscript to present the formulas in a chronological order, as suggested, to facilitate easier reference for readers. Additionally, we have reordered the abbreviations accordingly to ensure consistency and improve readability.

  1. The accepted abbreviation for alkaline phosphatase is “ALP”. Authors need to change the abbreviation throughout the manuscript.

Response: Thank you for your valuable feedback. We appreciate your suggestion regarding the abbreviation for alkaline phosphatase. As per your recommendation, we have updated the manuscript to consistently use "ALP" as the accepted abbreviation throughout the text.

  1. Table 3: Authors mentioned in the lines that there are some statistical significances among the diets. However, there was no representation of statistical significance in the table. Authors need to check the table. If there is no statistically significant, authors need to rephrase the results section.

Response: Thank you for your constructive feedback. We appreciate your attention to detail regarding the statistical significance mentioned in the results section and its representation in Table 3.Upon review, we have confirmed that there were no statistically significant differences among the diets in the data presented. We have therefore made the necessary revisions to Table 3 to ensure that it accurately reflects this, and we have also updated the results section to remove any reference to statistical significance.

  1. Figure 1: the bar columns of the figure were not clearly visible in the pdf. Authors need to check the figure 1.

Response: Thank you for your valuable feedback. We apologize for the visibility issue with Figure 1 in the PDF. We have reviewed and enhanced the quality of the figure by increasing the contrast and adjusting the resolution to ensure that the bar columns are clearly visible. We have also verified the revised figure’s clarity in the PDF format. Please find the updated Figure 1 in the revised manuscript.

Round 2

Reviewer 2 Report

Comments and Suggestions for Authors

The authors have adequately addressed the concerns previously raised through their revisions.

Author Response

We sincerely appreciate the reviewer’s positive feedback and acknowledgment of the revisions made. We are glad to know that the concerns have been adequately addressed. Should there be any further suggestions or questions, we are more than willing to make any additional improvements. Thank you for your constructive comments and valuable time.